# Analysis of the Anticipatory Behavior Formation Mechanism Induced by Methamphetamine Using a Single Hair

**DOI:** 10.3390/cells12040654

**Published:** 2023-02-17

**Authors:** Riku Sato, Megumi Kanai, Yukina Yoshida, Shiori Fukushima, Masahiro Nogami, Takeshi Yamaguchi, Norio Iijima, Kenneth Sutherland, Sanae Haga, Michitaka Ozaki, Kazuko Hamada, Toshiyuki Hamada

**Affiliations:** 1Department of Pharmaceutical Sciences, International University of Health and Welfare, Ohtawara 324-8501, Japan; 2Center for Basic Medical Research, International University of Health and Welfare, Ohtawara 324-8501, Japan; 3Global Center for Biomedical Science and Engineering, Hokkaido University, Sapporo 060-8012, Japan; 4Department of Biological Response and Regulation, Faculty of Health Sciences, Hokkaido University, Sapporo 060-0812, Japan; 5Hakujikai Institute of Gerontology, 5-11-1, Shikahama, Adachi Ward, Tokyo 123-0864, Japan

**Keywords:** circadian rhythm, *Period1*, in vivo, luciferin, methamphetamine, anticipatory behavior

## Abstract

While the suprachiasmatic nucleus (SCN) coordinates many daily rhythms, some circadian patterns of expression are controlled by SCN-independent systems. These include responses to daily methamphetamine (MAP) injections. Scheduled daily injections of MAP resulted in anticipatory activity, with an increase in locomotor activity immediately prior to the time of injection. The MAP-induced anticipatory behavior is associated with the induction and a phase advance in the expression rhythm of the clock gene *Period1* (*Per1*). However, this unique formation mechanism of MAP-induced anticipatory behavior is not well understood. We recently developed a micro-photomultiplier tube (micro-PMT) system to detect a small amount of *Per1* expression. In the present study, we used this system to measure the formation kinetics of MAP-induced anticipatory activity in a single whisker hair to reveal the underlying mechanism. Our results suggest that whisker hairs respond to daily MAP administration, and that *Per1* expression is affected. We also found that elevated *Per1* expression in a single whisker hair is associated with the occurrence of anticipatory behavior rhythm. The present results suggest that elevated *Per1* expression in hairs might be a marker of anticipatory behavior formation.

## 1. Introduction

Circadian physiology and behavioral rhythms are regulated in mammals by a central pacemaker within the suprachiasmatic nucleus (SCN) located in the hypothalamus. Destruction of the SCN disrupts the rhythms of many physiological functions. Environmental light is the strongest zeitgeber for the circadian system. The SCN controls the phase of daily rhythms and synchronizes circadian system responses to environmental light. It also coordinates peripheral tissue activity rhythms [1,2].

It has been reported that drugs such as methamphetamine (MAP) are capable not only of entraining the circadian system, but also of driving rhythms in the absence of the SCN [3,4,5,6]. Scheduled daily injections of MAP result in anticipatory activity, with an increase in locomotor activity immediately prior to the time of injection. This phenomenon has been documented in both rats and mice, and robust increases in activity levels can be observed following the expected time of injection, even on a day when the drug is withheld [7,8]. The MAP-induced change in the normal circadian pattern of activity is associated with a phase in the expression of the clock genes *Period1* (*Per1*) and *Per2* within the striatum and liver, but not the SCN [8]. Scheduled injections of MAP also reinstate behavioral circadian rhythms in arrhythmic SCN-lesioned animals [8]. These rhythms persist on withdrawal days and are associated with the reinstatement of circadian rhythms of *Per* expression in the striatum and liver of the SCN-lesioned animals [8]. These results suggest that the activity and clock gene expression rhythms driven by scheduled MAP injections are SCN-independent. Additionally, scheduled MAP injections are reported to shift the phase of peak clock gene expression in peripheral tissues around the time of the injection [8,9].

MAP-induced anticipatory activity is, at least in part, controlled by the dopaminergic system [10,11]. However, the formation mechanism of MAP-induced anticipatory activity is not well understood. As previously reported, MAP-induced elevated *Per1* expression is associated with anticipatory behavior. However, these data of MAP in the biological clock have been studied by measuring the clock gene expression rhythm of each tissue in the living body via sampling at specific times of the day, using methods such as polymerase chain reaction (PCR) and in situ hybridization, and analyzing gene expression from several animal tissues or using tissue culture after administering MAP [8,9]. With these methods, it was unclear when MAP-induced anticipatory activity developed after MAP injection. We previously reported that the *Per1* in the scalp and whisker hairs could be used as a marker of diabetic aggravation induced by the disruption of circadian clock gene expression [12,13]. In the present study, we examined whether the process of formation of MAP-induced anticipatory activity could be measured using simple and non-invasive methods to measure Per1 expression in the scalp and whisker hairs as a marker of the formation of anticipatory activity.

## 2. Materials and Methods

### 2.1. Animals

Mice were born and reared in our animal quarters. The environmental conditions were controlled as follows: 12 h light/12 h dark (LD) cycle with lights on at 8:00 and off at 20:00 (zeitgeber time (ZT) 0–12), temperature 23 ± 1 °C, and humidity 50 ± 5%. C57BL/6J mice carrying a *Per1*-promoter-driven firefly luciferase reporter gene (*Per1-luc*) were used [14]. The genetically modified mice used in this study and their handling methods were described in our previous study [13,14]. All animal work was performed in accordance with the Guidelines for the Care and Use of Laboratory Animals in the International University of Health and Welfare, with permission #18014 from the Committee for Animal Experimentation.

### 2.2. Methamphetamine Injection

Methamphetamine HCl (MAP) (Dainippon Pharmaceutical Co., Osaka, Japan), dissolved in saline (0.1 *w*/*v*%), was intraperitoneally injected at a fixed time (ZT3) on 5 consecutive days (MAP1 0–4). Then, following 2 days with no treatment, MAP was injected for another 5 consecutive days (MAP2 0–4). Injection-anticipatory activity was defined as activity occurring during the two hours prior to injection (ZT1-3). Injection-induced activity was defined as activity occurring during the 6 h following the injection. MAP was given to male and female *Per1-luc* mice (C57 BL/6J background). The effect of MAP on behavioral activity rhythms showed no significant differences between male and female mice [4].

### 2.3. Locomotor Activity Rhythm

*Per1-luc* mice were housed in transparent plastic cages. A chronobiology kit [12,13] was used for measuring their locomotor activity rhythms. Behavior records in 5-minute bins were used for this analysis. Food and water were provided ad libitum. The plastic cages and method of determining locomotor activity have been described previously [12,13].

### 2.4. Measurement of Abdominal Temperature

Intraperitoneal temperature was continuously recorded with a thermometer device (Thermochron Type-G, #1921G, KN Laboratories, Inc., Osaka, Japan) containing a thermometer, microchip, and battery. The device monitored temperature at 30 min intervals. The temperature resolution was 0.5 °C. Implantation of the device into the abdominal cavity was performed under isoflurane anesthesia. After the experiment was over, the device was taken out of the abdominal cavity, and the data were read using the provided software “ThermoManager (ver. 2.27)”.

### 2.5. Detection of Per1 Expression Rhythms in the Skin of Freely Moving Mice

To record *Per1* expression in the skin, mice were anesthetized with isoflurane (Zoetis, Tokyo, Japan). A tissue contact optical sensor (TCS) [15] was implanted under the skin of a *Per1-luc* mouse and fixed so that the photocathode was in close contact with the subcutaneous skin tissue, as previously reported [13]. After surgery, the mouse was housed in LD and allowed to establish stable daily activity rhythms. The lights-off time (20:00) was designated as ZT12. Optical fibers were then connected to the portable optical detection (POD) device [15,16]. The data from the POD were saved on an SD card and analyzed as previously reported [16].

The bioluminescence recording began in constant darkness (DD). The locomotor activity onset time was designated as circadian time (CT)12 under DD conditions. D-luciferin (20 mg/mL) (Fujifilm Wako, Osaka, Japan) was injected intraperitoneally into *Per1-luc* mice at a controlled flow rate (10 μL/h) as described in our previous report [17], with some modifications, in order to achieve sufficient photon counts. Photons emitted by the target areas of freely moving mice were integrated over 10-second intervals and averaged for 30 min during long recordings. Our method of determining the peak time of *Per1* expression rhythm was described previously in [15].

### 2.6. Detection of MAP-Induced Per1 Expression in the Scalp Hairs Using Micro-PMT

MAP (0.1 *w*/*v*%, 0.5 mL) was intraperitoneally injected into adult *Per1-luc* mice at ZT3 (AM 11:00). Food and water were provided ad libitum. Before MAP injection (day 0), and after 1 day (day 1), 2 days (day 2), 3 days (day 3), 4 days (day 4), and 7 days (day 7), mouse scalp hairs were collected. Mouse scalp hairs (over a dozen hairs) were isolated by forceps at ZT3 and incubated in 100 μL of PicaGene cell culture reagent (Toyo B-net Co., Ltd., Tokyo, Japan) over several hours. After centrifugal spinning, the supernatant (25 μL) was mixed with 50 μL of PicaGene (Toyo B-net Co., Ltd.). The bioluminescence of the mixed solution (25 μL) was measured every 10 s using a micro-PMT system as per our previous methods [13]. The supply voltage was set to 800 V in order to obtain bioluminescence. The value of bioluminescence was calculated using the data measured from 120 s to 300 s, to eliminate instability in the initial measurement of the micro-PMT. After the measurement, the number of scalp hairs was counted. The value of bioluminescence was calculated per hair.

### 2.7. Detection of MAP-Induced Per1 Expression in a Whisker Hair Using Micro-PMT

#### 2.7.1. Solution Method

Similar to our experiment examining scalp hair, we pulled a whisker hair out for the experiments to determine MAP-induced *Per1* expression in a single hair. The sample was incubated in 25 μL of PicaGene cell culture reagent over several hours. After centrifugal spinning, the supernatant (25 μL) was mixed with 50 μL of PicaGene. The bioluminescence of the mixed solution (25 μL) was measured every 10 s using a micro-PMT system as per our previous methods [13]. The supply voltage was set to 1100 V to order to obtain sufficient bioluminescence. The value of bioluminescence was calculated using the data measured from 120 s to 300 s, to eliminate instability in the initial measurement of the micro-PMT.

#### 2.7.2. Direct Methods

A whisker hair was pulled out with the same solution method described above. Within 10 min after sampling, the sample was placed directly on the photocathode of the micro-PMT, which was prefilled with PicaGene solution (5 μL). The bioluminescence was measured every 10 s for 300 s within 10 min after sampling. The supply voltage was set to 1100 V in order to obtain sufficient bioluminescence. The value of bioluminescence was calculated using the data measured from 120 s to 300 s, to eliminate instability in the initial measurement of the micro-PMT. Two methods were tested: measuring the amount of luminescence from a hair cut in half by micro-PMT, and measuring without cutting.

### 2.8. LUC Expression in Scalp and Whisker Hairs by Immunohistological Analysis

The expression of *Per1* in scalp and whisker hairs was immunohistochemically identified in *Per1-luc* mice using an anti-luciferase polyclonal antibody (Promega, Madison, WI, USA). Scalp hairs and a single whisker hair were pulled out at ZT3 and stored in cold 4% paraformaldehyde in 0.1 M phosphate buffer (pH 7.4). Fixed samples were kept in phosphate buffer before the immunostaining experiment. The method of determining LUC expression has been described previously [13,18].

### 2.9. Statistics

The effects of MAP treatments on *Per1* expression and the period of locomotor activity rhythm were examined using either one-way ANOVA followed by Dunnett’s test or two-way ANOVA followed by Bonferroni’s multiple comparisons test.

## 3. Results

### 3.1. Effect of Daily MAP Injection on Locomotor Activity and Body Temperature Rhythm

Figure 1A,B show that locomotor activity was repeated according to the scheduled injection of MAP (0.1 *w*/*v*%, 0.5 mL i.p.). *Per1-luc* mice were injected with MAP at ZT3 for 5 days (MAP1 0–4) and withdrawn for 2 days (WD 1–2) following the 5 days of injections (MAP2 0–4). MAP injection resulted in an increase in daily activity occurring from ZT5-9, 2–6 h after the injections. Before MAP injection, the *Per1-luc* mice were active at night and resting during the day. Immediately after MAP administration, the mice showed the reverse, resting at night and being active during the day (Figure 1C). This injection-induced activity disappeared immediately after withdrawing MAP (WD1 1–2). With these reversed conditions, we examined whether physiological functions such as glucose content, body weight, and water intake were affected. Appendix A shows that these functions had no change if daily MAP injections continued for 7 days.

Anticipatory activity was observed after MAP injection on withdrawal days (WD1 1–2 and WD2 1–5) (red square areas in Figure 1A). Figure 1D shows the locomotor activity before MAP injection (pre1–4 days), during MAP injection, and on withdrawal days 1 and 2. Locomotor activity significantly increased at day 3 and day 4 after MAP injection. Withdrawal days 1 and 2 also showed a significant increase in locomotor activity. These results suggest that anticipatory activity is formed within 3 days after MAP injections. During this duration, the saline-treated group had no increase in locomotor activity (mean % of daily activity for the whole duration: 0.83 ± 0.35). Figure 1E shows that anticipatory activity continued for at least 9 days after the withdrawal of MAP.

Next, we examined the effect of MAP on body temperature rhythm (Figure 1F–H). The peak of pre-feeding activity rhythm by restricted daily feeding (RF) in the light phase was previously determined to develop body temperature rhythm [19]. Before MAP injection, the peak time of body temperature rhythm was 5:08 ± 0.38 h (lights on 8:00, lights off 20:00 in the LD cycle). On the first and second days of MAP injections, there were two peaks of body temperature. Peak body temperature showed a stable rhythm on the 3rd day of MAP injection. The mean peak time (MAP1 2–4) was 19:00 ± 0.56 h at MAP1 2–4. Withdrawal of MAP (WD 1–1, 1–2) immediately abolished the peak of body temperature rhythm, and two peaks appeared. When MAP was re-administered (MAP2 0–4), the peak body temperature rhythm immediately shifted to the end of the light phase in the LD cycle on the first day of MAP injection. It showed a stable rhythm during MAP2 0–4. The mean peak time was 18:27 ± 1.46 h. These peaks disappeared again after the withdrawal of MAP (WD 2–1, 2–2). This phenomenon was thought to be due to the reverse tolerance brought about by daily MAP administration [20]. After 3 days of withdrawal, the peak body temperature rhythms were immediately set to the dark phase in the LD cycle (peak time was 3:20 ± 0.12 h) compared to that of MAP1 0–4.

We also analyzed the amplitude of body temperature rhythm (Figure 1H). The decrease in amplitude was observed at the MAP (1–0, 1–1), WD (1–1, 1–2), and WD (2–1, 2–2). After the decrease in the amplitude of body temperature rhythm at the time when body temperature rhythm had two peaks, the body temperature showed a stable rhythm. The MAP-injection-induced shift in body temperature rhythm did not continue in the same manner as the behavior activity rhythm.

### 3.2. MAP-Induced Per1 Gene Expression in the Scalp Hairs

We previously reported that elevated *Per1* expression caused by streptozotocin (STZ) plays an important role in the aggravation of diabetes, and that *Per1* expression in the back skin hairs responded to blood glucose changes in very early stages of diabetes [12,13]. These results suggest that the experimental procedure using hairs is a simple and easy measure to detect diseases. Therefore, using the measurement of *Per1* expression in hairs, we attempted to examine whether *Per1* expression changes associated with MAP-induced anticipatory activity were detected.

We first confirmed the *Per1* expression rhythm in the skin (including the hairs) on the first day (DD day1) after transferring mice from LD to DD (Figure 2A), in order to determine the sampling time of hairs of *Per1-luc* mice. We did this because, under LD conditions, LD light affects the measurement of *Per1* gene expression in the skin, so it is necessary to measure the *Per1* expression rhythm under DD conditions. In Figure 2A, the *Per1* expression rhythm in the back skin is shown at DD day 1 and DD day 2. The peak time of *Per1* expression at DD day 1 was CT11.4 ± 2.01 h (*n* = 4 animals). It is reported that the PER2 protein is expressed in most cell types, including epidermal keratinocytes, dermal fibroblasts, and hair follicles of the skin. In these cells, the PER2 protein is expressed with a peak at CT16, 4–8 h after expression of its mRNA [21]. Then, we estimated *Per1* expression in the scalp hairs and a whisker hair during the time period of low *Per1* expression; it was easy to detect the effect of MAP on *Per1* expression.

A schematic figure of scalp hair sampling at ZT3 is shown in Figure 2B. Mouse scalp hairs were collected before MAP injection (day 0) and then 1 day (day 1), 2 days (day 2), 3 days (day 3), 4 days (day 4), and 7 days (day 7) after injection. As previously reported, the levels of *Per1* expression in the back skin, liver, olfactory bulb, cortex, and ear at ZT3 were low [12,13,17]. After we determined that *Per1* was strongly expressed in scalp hair follicles after MAP injection (Figure 2C), changes in *Per1* expression in the scalp hairs after MAP injection were examined. Elevated *Per1* expression was measured from the bioluminescence of the scalp hair samples (Figure 2D), as previously reported [12,13]. Elevated *Per1* expression in the scalp hair samples was detected at days 4 and 7. After a two-day MAP withdrawal period, significant *Per1* expression was detected on day 7 and day 4. *Per1* expression on day 3 tended to be increased, despite the present studies showing no significant value. These results suggest that *Per1* expression rhythm in scalp hairs was reset at least four days after daily MAP injection.

### 3.3. MAP-Induced Per1 Gene Expression in a Whisker Hair

Similarly, we attempted to detect *Per1* expression in a single whisker hair, with the same schedule as shown in Figure 2B. We confirmed that *Per1* was strongly expressed in the whisker hair follicle (Figure 3A). To quantify the elevation of *Per1* expression caused by MAP injection, we prepared a sample solution of cell extracts from a randomly collected whisker hair to react with a PicaGene (Toyo B-net Co., Ltd.) solution containing D-luciferin as a substrate. The *Per1* expression level was quantified by measuring the amount of bioluminescence by placing the sample solution on the photocathode [13] of the micro-PMT (Figure 3B). We detected a significant increase in *Per1* expression in the whisker hair at day 3, day 4, and day 7 after MAP injection (Figure 3C). After a two-day MAP withdrawal period, significant *Per1* expression was detected on day 7, as well as on day 3 and day 4 (Figure 3C).

Next, we tried to determine the elevation of *Per1* expression more simply than the above method using a “sample solution”. We collected a whisker hair randomly and placed it directly on the photocathode of the micro-PMT, which was filled with PicaGene solution (Figure 3D). Like the cell solution method, we detected a significant increase in *Per1* expression in the whisker hair at day 3, day 4, and day 7 after MAP injection (Figure 3E). We named this method the “direct method”. The direct method was very simple because it involved placing the hairs on the micro-PMT. Interestingly, we could detect *Per1* expression in whisker hair cells by simply mixing a whisker hair with PicaGene solution. Five minutes was enough to analyze the *Per1* expression in whisker hair cells. The measurement of bioluminescence by the luciferin–luciferase reaction to detect *Per1* expression in whisker hair cells requires that the luciferin in the PicaGene solution reacts with the luciferase within the whisker hair. Therefore, we compared whether the root area of a whisker hair that was cut in half showed higher bioluminescence than that of an uncut hair. Appendix A shows that the direct method (uncut sample) was more sensitive to measure *Per1* expression than the half-cut sample whisker hair (Bioluminescence values, uncut with no PicaGene: 142.5 ± 95.2; uncut with PicaGene: 193,180.8 ± 90,545.5; cut with PicaGene: 67,316.1 ± 46,957.8).

## 4. Discussion

We succeeded in detecting elevated *Per1* expression in a single whisker hair as a result of daily MAP injections. As a direct effect of MAP, an increase in activity was observed from the first day of MAP administration. This direct effect occurred between 2 and 6 h after MAP administration. The increase in activity disappeared 6 h after MAP administration (Figure 1B). This increase in activity occurred only during the period of MAP administration and did not occur without MAP administration. It has been reported that the serum half-life of MAP after oral administration is about 8.5 h [22]. We measured *Per1* expression 24 h after MAP administration in this study. The luciferase reaction was used to measure *Per1* expression, because the half-life of LUC is shorter (2–3 h) than that of MAP [23,24]. Therefore, 24 h after MAP administration, the direct effect of MAP is thought to have disappeared. *Per1* expression near the root of the hair increased significantly 3 days after MAP administration, but not during day 1 and day 2 after MAP administration. This increase was measured for up to 7 days without MAP administration. Based on these results, this increase cannot be attributed to a direct effect of MAP. Considering the elimination of the direct effect of MAP on *Per1* expression after MAP administration, the present results suggest that elevated *Per1* expression in hairs can be used as a marker of anticipatory behavior formation. These results also show an increase in sensitization (Figure 1C–H and Figure 3C,E) to MAP after MAP injections [20]. Daily MAP injections at a fixed time resulted in an anticipatory activity rhythm in *Per1-luc* mice with a C57BL/6 background that persisted on withdrawal days, as previously reported [8,9]. Body temperature rhythms shifted to a new phase within 3 days after daily MAP injections, but these rhythms had no continuity. Circadian changes in temperature comparable to those seen in core body temperature rhythms (36 °C to 38.5 °C in mice) can entrain and enhance the amplitude of circadian rhythms in peripheral tissues [25]. In the present study, *Per1* expression in scalp and whisker hairs maintained high levels on withdrawal days. This elevated *Per1* expression indicates that it is not a result induced by the change in body temperature rhythm. It has been reported that daily oscillation of *Period* genes outside the SCN (striatum, pituitary, salivary gland, and liver) is closely associated with the regulation of SCN-independent rhythms similar to the MAP-induced anticipatory behavior rhythm [8,9]. The present study also provides the first demonstration that hairs are affected by repeated MAP injections.

The skin is a large and complex organ composed of multiple cell types, organized into layers, featuring thousands of mini-organ structures such as hair follicles and sweat glands. Circadian oscillations were found to be present in several principal skin cell types, including epidermal and hair follicle keratinocytes, dermal fibroblasts, and melanocytes [26,27,28,29,30]. At least 1400 genes involved in multiple functions show circadian expression changes in mouse skin [31]. MAP-induced anticipatory behavior is associated with the induction and a phase advance in the expression rhythm of *Per1* [8]. CCD camera and TCS studies show that *Per1* expression in the skin of freely moving mice has a peak around the time of activity onset [13,17]. Therefore, we determined the time (ZT3) of MAP injection by considering *Per1* expression rhythms in the skin. At ZT3—the time of MAP administration—*Per1* expression was found to be low (Figure 2A). As a result, it was easy to detect the changes in *Per1* expression induced by MAP injections. In other papers [8,9], MAP was injected at ZT7, and anticipatory behavior was induced by MAP injection, as in our results. Using this time point (ZT3), we observed elevated *Per1* expression in scalp and whisker hairs as a result of the MAP injections. Significant *Per1* expression in scalp hairs was induced after MAP injection at day 4. In addition, the *Per1* expression was maintained at day 7 when MAP was withdrawn. Similarly, we showed that significant *Per1* expression in a single whisker hair was induced after MAP injection at days 3 and 4 and continued until day 7, when MAP was withdrawn. Considering the reports that the MAP-induced anticipatory behavior is associated with the induction and a phase advance in the expression rhythm of *Per1* [8], the present results suggest that the MAP-induced anticipatory system forms within 3 days of MAP administration. There was a difference in the *Per1* expression induced by MAP between scalp hairs and whisker hairs. Whisker hairs are longer, stiffer, and larger in diameter than scalp hairs. In addition, whisker hairs are innervated to the somatosensory cortex, where a column of neurons called “barrels” is widely used as a model system to study cortex development, neuronal plasticity, and sensory motor integration [32]. Whisker hairs may play some role in the MAP-induced anticipatory behavior formation mechanism through the cortical barrel. We have reported that STZ treatment induces different *Per1* expression in scalp and whisker hairs. Further studies are needed to clarify the difference between scalp and whisker hairs. Finally, we developed a system for analyzing *Per1* expression by direct sampling. The direct method is very simple, and its sensitivity was enough to detect the *Per1* expression in a single whisker hair. This method can likely be applied to various other biological studies.

At present, the mechanism underlying the anticipatory expression of *Per1* in hairs in response to daily injection with MAP remains unknown. The circadian clock activity in the skin and hair is coordinated by the SCN, presumably through neuronal and hormonal mediators, under normal conditions. In MAP-induced conditions, an unknown region of the brain regulates the circadian clock activity of hairs. This is thought to be under the control of the SCN [33], and it might be the same regulating activity outside the SCN (striatum, pituitary, salivary gland, liver, or hair) and anticipating behavior to activate the dopamine D1 receptor or NMDA receptors [9,11]. Further whole-body studies of the target area formed by activating MAP-induced anticipatory activity need to be conducted.

Several results have been reported for applying the method of analyzing clock gene expression in hair tissue to humans. PCR or ex vivo culture of hair tissues might be available [34,35]. Applying the present method of hair tissue to healthy and MAP-treated subjects may lead to an understanding of the relationship between the impact of drug abuse and the disturbance of circadian rhythms.

## Figures and Tables

**Figure 1 cells-12-00654-f001:**
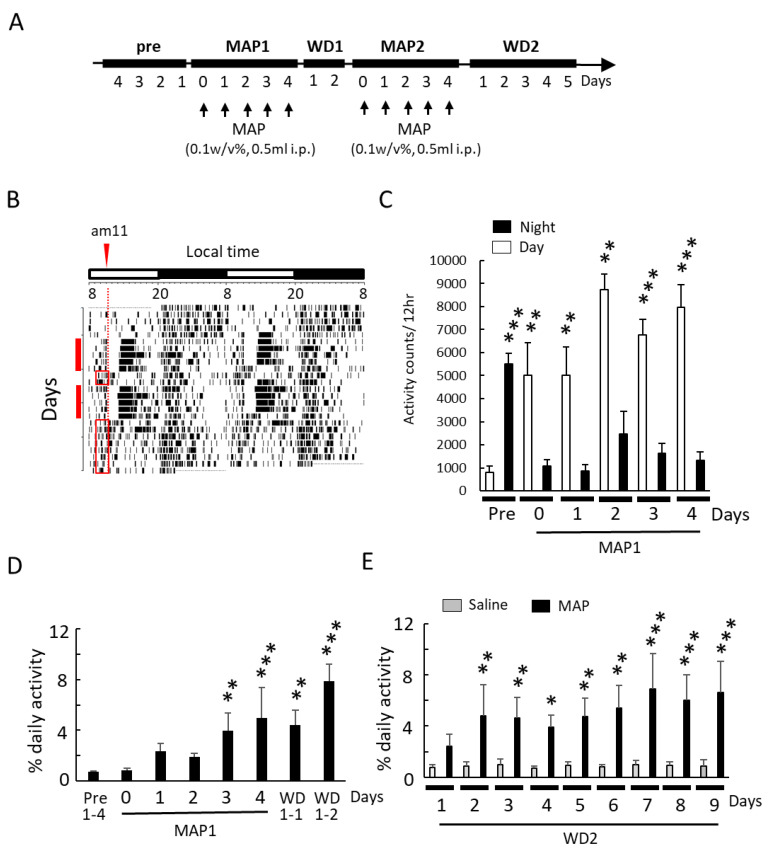
Effect of daily MAP injections on locomotor activity and body temperature rhythm. (**A**) Schematic figure of MAP intraperitoneal (i.p.) injection schedule at ZT3. (**B**) Anticipatory locomotor activity. Representative double-plotted actographs of the *Per1-luc* mice subjected to MAP are shown. The red line in the vertical bar indicates the time of injection (MAP1 0–4, MAP2 0–4) at ZT3 (AM11:00). Levels of behavioral activity are indicated by black columns. The figure depicts locomotor activity beginning prior to the start of injections. MAP injection resulted in an acute increase in locomotor activity, with anticipatory activity prior to the scheduled injection, and an increase in activity following injection. Red boxes indicate anticipatory activity without MAP injections. The horizontal bar at the top of each panel represents the light cycle—the light phase with open bars and the dark phase with filled bars. (**C**) The amount of locomotor activity during the day and night. After MAP injection, the amount of locomotor activity during the day and night was reversed. “Pre” indicates before MAP injection. Day 0 indicates the start day of MAP injection. Open and closed columns show day (08:00–20:00) and night (20:00–08:00), respectively. Statistical significance was determined by one-way ANOVA followed by Dunnett’s test (*n* = 4 animals) (** *p* < 0.01, *** *p* < 0.001 vs. Pre Day). (**D**) Amount of locomotor activity between 9:00 and 11:00 at Pre, MAP1 0–4, WD1–1, and WD1–2. MAP was withdrawn at WD1–1 and WD1–2 after daily MAP injection at ZT3 (11:00) for 5 days (MAP1 0–4). Vertical values show the anticipatory activity (%) {[(activity counts from 2 h)/(activity counts for a full day)] × 100} on the day before or after MAP. Statistical significance was determined by one-way ANOVA followed by Dunnett’s test (*n* = 4 animals) (** *p* < 0.01, *** *p* < 0.001 vs. Pre 1–4). (**E**) Anticipatory activity persisted after withdrawal. MAP was withdrawn after daily MAP injection at ZT3 (11:00) (MAP2 0–4). Day 1 indicates the day after withdrawing MAP. Statistical significance was determined by two-way ANOVA followed by Bonferroni’s multiple comparisons test (*n* = 4 animals) (* *p* < 0.05, ** *p* < 0.01, *** *p* < 0.001 vs. saline). (**F**) Body temperature rhythm of the same mouse in Figure 1A. Body temperature of the *Per1-luc* mice subjected to MAP is shown. The body temperature is plotted at 30 min intervals. Red lines indicate fitted curves for each plot. The bar in the vertical axis indicates the time of injection (MAP1 0–4, MAP2 0–4) at ZT3 (11:00). (**G**) Peak time of body temperature rhythm (*n* = 3 animals). The peak time at MAP1–0, MAP1–1, WD1–1, WD1–2, WD2–1, and WD2–2 was not clear. These peak times indicate two peaks. (**H**) Amplitude of body temperature rhythm (*n* = 3 animals).

**Figure 2 cells-12-00654-f002:**
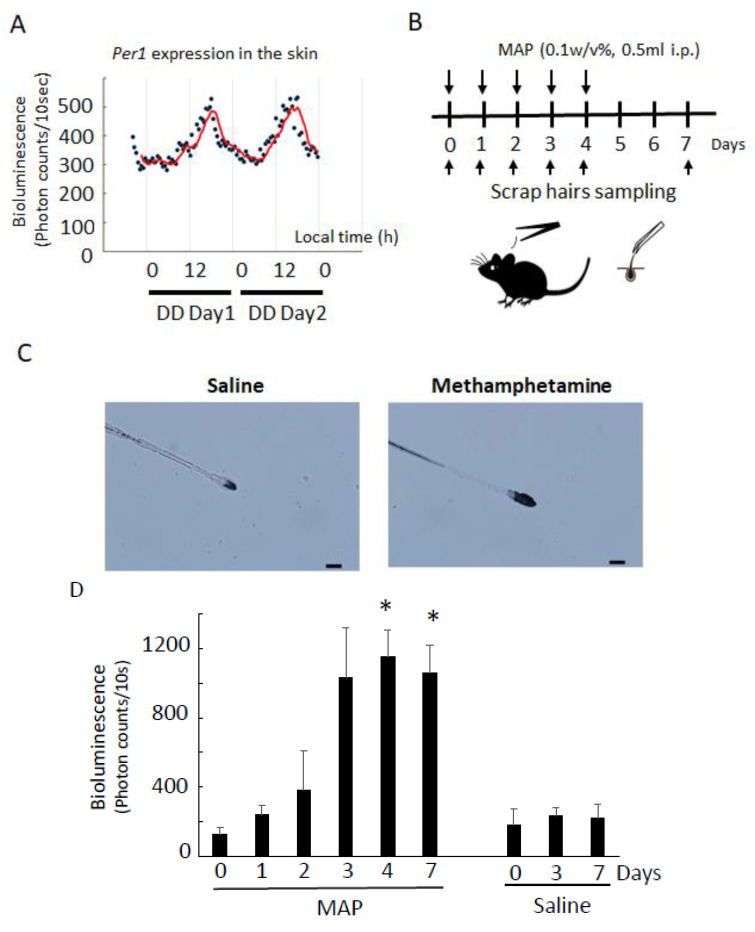
MAP-induced *Per1* gene expression in scalp hairs. (**A**) *Per1* bioluminescence (photon counts/10 s) of the skin in a freely moving *Per1-luc* mouse with a TCS [15]. The TCS was implanted under the skin of the *Per1-luc* mouse. Bioluminescence was measured from DD day 1. Photons emitted by the skin cells of freely moving mice were integrated over 10 s intervals and averaged for 30 min. The red line is the trend line. (**B**) Schematic figure of scalp hair sampling and MAP injections at ZT3. Day 0 indicates the start day of MAP injections. Sampling of the scalp hairs was performed just before MAP injections. Mouse scalp hairs were isolated by forceps and incubated in PicaGene cell culture reagent until the analysis using a micro-PMT system. (**C**) Expression of *Per1* in scalp hair. The hairs were sampled at ZT3 on day 7. The photos indicate the immunostaining with LUC polyclonal antibody. The scale bar is 200 μm. (**D**) MAP-induced *Per1* gene expression in scalp hair after daily injection of MAP. Bioluminescence was measured every 10 s using a micro-PMT system [13]. The supply voltage was set to 800 V in order to obtain the bioluminescence. Statistical significance was determined by one-way ANOVA followed by Dunnett’s test *(n =* 4 animals) (* *p* < 0.05 vs. day 0). The day 0 sample was collected just before MAP injection.

**Figure 3 cells-12-00654-f003:**
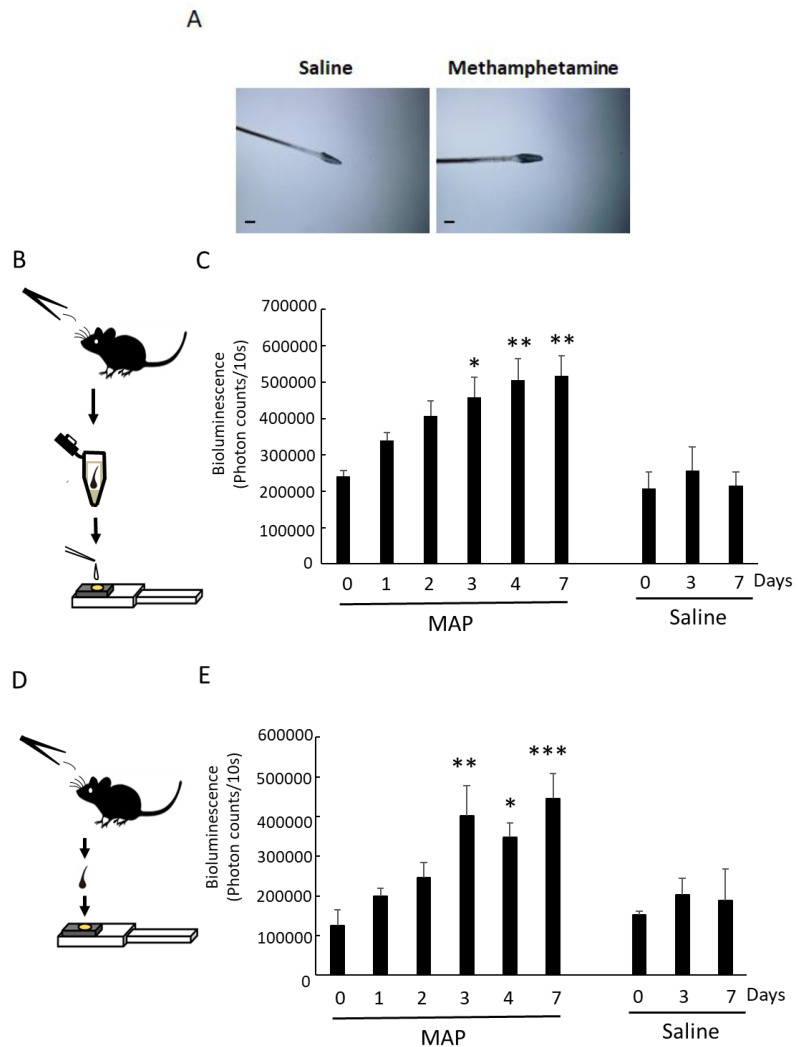
MAP-induced *Per1* gene expression in whisker hairs. (**A**) Expression of *Per1* in whisker hairs. The hairs were sampled at ZT3 on day 7. The photos indicate the immunostaining with LUC polyclonal antibody. The scale bar is 200 μm. (**B**) Schematic figure of the solution method for measuring *Per1* expression in whisker hairs. (**C**) MAP-induced *Per1* gene expression in whisker hairs, determined by the solution method. Bioluminescence was measured every 10 s using a micro-PMT system [13]. The supply voltage was set to 1100 V in order to obtain sufficient bioluminescence. Statistical significance was determined by one-way ANOVA followed by Dunnett’s test (*n* = 4 animals) (* *p* < 0.05, ** *p* < 0.01 vs. day 0). The day 0 sample was collected just before MAP injection. (**D**) Schematic figure of the direct method for measuring *Per1* expression in whisker hairs. (**E**) MAP-induced *Per1* gene expression in whisker hairs, determined by the direct method. Statistical significance was determined by one-way ANOVA followed by Dunnett’s test (*n* = 4 animals) (* *p* < 0.05, ** *p* < 0.01, *** *p* < 0.001 vs. day 0). The day 0 sample was collected just before MAP injection.

## Data Availability

Data are contained within the article.

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
