# Peer review of "Analysis of the Anticipatory Behavior Formation Mechanism Induced by Methamphetamine Using a Single Hair"

_cells, 2023, doi:10.3390/cells12040654_

Round 1
Reviewer 1 Report
Major comments: This is a lovely technique and a possibly interesting finding. The technique (method) is convincing, but the experimental results less so.
The authors do not show an actogram of anticipatory locomotor activity after the first few days of MAP treatment. This information must be quantified for the group of animals- not just in an actogram of one animal. To say it another way, there is no quantification of anticipatory activity. All the quantification of activity appears to be for post injection time slots. Hour by hour activity plots would be the most transparent way to showing the results.
I am not sure that the authors show what is claimed, though possibly I issed something. The second sentence of discussion says that anticipation may occur at day 3. This appears to be based on increased bioluminescence on day 3 compared to day 0-2. However, the authors should show some control data for Per1 bioluminescence. e.g. Is there data available from saline-treated animals? Also- hopefully there is data available for bioluminescence at non-anticipatory times of day. In the absence of data comparing anticipatory and non-anticipatory time slots, it is not possible to interpret the results as an increase of bioluminescence in anticipation. It might be that continued MAP treatment augments Per1 expression.
Minor comments:
There are numerous typos that can be readily corrected.
eg as a maker of the formation of antici- 72
Representative double-plotted autographs are shown of the Per1-luc mice subjected to 201
Fig 2 Legend: 278 scalp hairs were performed before MAP injections”. How long before injections were the samples taken?
There are numerous errors of English grammar. Perhaps the author could ask for assistance from a native English speaker.
Author Response
22 November 2022
Dear Sir:
Here we submit our revised manuscript entitled ` Analysis of anticipatory behavior formation mechanism induced by methamphetamine using a single hair’ authored by Riku Sato, Megumi Kanai, Yukina Yoshida, Shiori Fukushima, Masahiro Nogami, Takeshi Yamaguchi, Norio Iijima, Kenneth Sutherland, Sanae Haga, Michitaka Ozaki, Kazuko Hamada, Toshiyuki Hamada, which we send back to you for possible publication in ‘cells’.
We greatly appreciate the comments provided to our paper. Our responses to the comments are summarized below.
Reviewer’ comments and our responses:
Comment1: The authors do not show an actogram of anticipatory locomotor activity after the first few days of MAP treatment. This information must be quantified for the group of animals- not just in an actogram of one animal. To say it another way, there is no quantification of anticipatory activity. All the quantification of activity appears to be for post injection time slots. Hour by hour activity plots would be the most transparent way to showing the results.
Our responses: Thank you for your comment. We added the data showing an actogram of anticipatory locomotor activity (%) the day before or after of after MAP in Figure1D and 1E.
Comment2: I am not sure that the authors show what is claimed, though possibly I issed something. The second sentence of discussion says that anticipation may occur at day 3. This appears to be based on increased bioluminescence on day 3 compared to day 0-2. However, the authors should show some control data for Per1 bioluminescence. e.g. Is there data available from saline-treated animals? Also- hopefully there is data available for bioluminescence at non-anticipatory times of day. In the absence of data comparing anticipatory and non-anticipatory time slots, it is not possible to interpret the results as an increase of bioluminescence in anticipation. It might be that continued MAP treatment augments Per1 expression.
Our responses: Thank you for your comment. We added the data in the Figure2D, Figure3C, 3E.
Minor comments: There are numerous typos that can be readily corrected. eg as a maker of the formation of antici- 72. Representative double-plotted autographs are shown of the Per1-luc mice subjected to 201
Our responses: Thank you very much for your comment. We corrected it.
Minor comments: Fig 2 Legend: 278 scalp hairs were performed before MAP injections”. How long before injections were the samples taken?
Our responses: Thank you very much for your comment. We added “just” before MAP injections.
Minor comments: There are numerous errors of English grammar. Perhaps the author could ask for assistance from a native English speaker.
Our responses: Thank you very much for your comment. Native English speaker checked the MS.
Reviewer 2 Report
The manuscript describes that methamphetamine-induced anticipatory behavior and Per 1 expression are analyzed using a single hair. The contents are quite interesting. I think the manuscript is worthy for publication in Cells. However, some revision is necessary as follows.
The authors injected methamphetamine at ZT 3. How did they decide the time? And then, if the injection time is different, how do the results change? They had better describe them.
The sensitization seems to be strengthened day by day on locomotor activity and bioluminescence (Fig 1 E, 3C and E). The authors should discuss it.
Title of 3.3 and Figure 3
Anticipatory locomotor activity is not described. So, “anticipatory locomotor activity” should be deleted.
L. 365~370
The different results between scalp hairs and whisker ones are little discussed. Whisker hairs may play important roles for sensory part. So, it might be able to discuss as an item of the reason. How about do the authors discuss about it?
There are some typographical errors.
L. 262~263
The level of Per 1 expression in the back skin, liver, olfactory bulb, cortex and ear at ZT# was lows we previously reported.
・・・Before “we” , “as” should be added?
L. 292
Per 1expression
Space is necessary between 1 and expression.
Author Response
22 November 2022
Dear Sir:
Here we submit our revised manuscript entitled ` Analysis of anticipatory behavior formation mechanism induced by methamphetamine using a single hair’ authored by Riku Sato, Megumi Kanai, Yukina Yoshida, Shiori Fukushima, Masahiro Nogami, Takeshi Yamaguchi, Norio Iijima, Kenneth Sutherland, Sanae Haga, Michitaka Ozaki, Kazuko Hamada, Toshiyuki Hamada, which we send back to you for possible publication in ‘cells’.
We greatly appreciate the comments provided to our paper. Our responses to the comments are summarized below.
Reviewer’ comments and our responses:
Comment1: The authors injected methamphetamine at ZT 3. How did they decide the time? And then, if the injection time is different, how do the results change? They had better describe them.
Our responses: We added the sentences in Discussion part at 434-439 lines. In the other papers (Iijima et al. EJN, 2002; Mohawk et al., PLoS One 2013), MAP was injected at ZT7, anticipatory behavior was induced by MAP injection as like our results.
Comment2:The sensitization seems to be strengthened day by day on locomotor activity and a bioluminescence (Fig 1 E, 3C and E). The authors should discuss it.
Our responses: We added the sentence in Discussion part at 421 line.
.Comment3:Title of 3.3 and Figure 3
Anticipatory locomotor activity is not described. So, “anticipatory locomotor activity” should be deleted.
Our responses: Thank you very much for your comment. We deleted the “anticipatory locomotor activity”
Comment4:L. 365~370: The different results between scalp hairs and whisker ones are little discussed. Whisker hairs may play important roles for sensory part. So, it might be able to discuss as an item of the reason. How about do the authors discuss about it?
Our responses: We added the sentences in Discussion part at 445-449 lines.
Comment5:There are some typographical errors.
- 262~263The level of Per 1 expression in the back skin, liver, olfactory bulb, cortex and ear at ZT# was lows we previously reported.
・・・Before “we” , “as” should be added?
Our responses: Thank you very much for your comment. We added “as” in the sentence.
Comment6:L. 292
Per 1expression
Space is necessary between 1 and expression.
Our responses: Thank you very much for your comment. We corrected it.
Round 2
Reviewer 1 Report
The authors have missed my major point. The response of the whiskers to Per1 IS INDEED seen by the third or 4rth day after Meth treatment. As is clear from the data shown here (and in the literature), anticipatory behavior develops several days later (no problem with all that). However, the authors have not shown, as far as I can tell, that the whisker response ANTICIPATES the Meth treatment by day 3-4. This can be addressed in a rewrite of the conclusions or additional experiments.Author Response
Comment: The authors have missed my major point. The response of the whiskers to Per1 IS INDEED seen by the third or 4rth day after Meth treatment. As is clear from the data shown here (and in the literature), anticipatory behavior develops several days later (no problem with all that). However, the authors have not shown, as far as I can tell, that the whisker response ANTICIPATES the Meth treatment by day 3-4. This can be addressed in a rewrite of the conclusions or additional experiments.
Our responses: Thank you for your comment. We rewrote the MS.
Round 3
Reviewer 1 Report
Please see the attached file.

Author Response
Comment1: The elevated Per1 expression in a single whisker hair is associated with the occurrence of anticipatory behavior rhythm. The present results suggest that elevated Per1 expression in hairs can be used as a maker of anticipatory behavior formation. THE KEY QUESTION IS WHETHER THE HAIR RESPONSE IS DIRECTLY DUE TO SEVERAL DAYS OF METH ADMINISTRA-TION OR WHETEHR THE ELEVATED PER1 OCCURS WITHIN THE 3 DAY TIME FRAME IN WHICH ANTICIPATORY ACTIVITY IS EXPRESSED. To distinguish between direct effects of meth on hair expression of per1 and the occurrence of antic-ipatory per1 expression on ~day 3, the authors need hourly data (similar to the behavioral data shown in FIG 3B). The authors can modify their conclusion to make it clear that these two possibilities cannot be distinguished. Alternatively, they can show hourly measurements of hair per1.
Our responses: We explained present results in revised MS to answer Reviewer's comment. We have included explanatory text that answers the reviewer’s question at the beginning of the Discussion part. We also replaced A and B in Figure 2 and rewrote the paper to make it easier to understand. In addition, we have English grammar checks by native speakers.